# Cannabidiol as Self-Assembly Inducer for Anticancer Drug-Based Nanoparticles

**DOI:** 10.3390/molecules28010112

**Published:** 2022-12-23

**Authors:** Eleonora Colombo, Davide Andrea Coppini, Laura Polito, Umberto Ciriello, Giuseppe Paladino, Mariafrancesca Hyeraci, Maria Luisa Di Paolo, Giulia Nordio, Lisa Dalla Via, Daniele Passarella

**Affiliations:** 1Department of Chemistry, Università degli Studi di Milano, 20133 Milan, Italy; 2Ann Romney Center for Neurologic Diseases, Department of Neurology, Brigham and Women’s Hospital and Harvard Medical School, Boston, MA 02115, USA; 3Department of Molecular Sciences, Institute of Natural Products and Agrobiology (IPNA), CSIC, 38206 La Laguna, Spain; 4Istituto di Scienze e Tecnologie Chimiche (SCITEC) “Giulio Natta”, Consiglio Nazionale delle Ricerche (CNR), 20138 Milan, Italy; 5LINNEA SA, 6595 Riazzino, Switzerland; 6Department of Pharmaceutical and Pharmacological Sciences, University of Padova, 35131 Padova, Italy; 7Department of Molecular Medicine, University of Padova, 35131 Padova, Italy; 8Consorzio Interuniversitario Nazionale per la Scienza e la Tecnologia dei Materiali (INSTM), 50121 Firenze, Italy

**Keywords:** self-assembled nanoparticles, cannabidiol, anticancer drug

## Abstract

Cannabidiol (CBD) is a biologically active compound present in the plants of the *Cannabis* family, used as anticonvulsant, anti-inflammatory, anti-anxiety, and more recently, anticancer drug. In this work, its use as a new self-assembly inducer in the formation of nanoparticles is validated. The target conjugates are characterized by the presence of different anticancer drugs (namely *N*-desacetyl thiocolchicine, podophyllotoxin, and paclitaxel) connected to CBD through a linker able to improve drug release. These nanoparticles are formed via solvent displacement method, resulting in monodisperse and stable structures having hydrodynamic diameters ranging from 160 to 400 nm. Their biological activity is evaluated on three human tumor cell lines (MSTO-211H, HT-29, and HepG2), obtaining GI_50_ values in the low micromolar range. Further biological assays were carried out on MSTO-211H cells for the most effective NP **8B**, confirming the involvement of paclitaxel in cytotoxicity and cell death mechanism

## 1. Introduction

Nanotechnology is an increasingly interesting approach in medicinal chemistry, allowing to improve the bioavailability and, thus, the delivery of drugs to their site of action. Among all types of nanoparticles (NPs), we have been primarily interested in lipidic, self-assembled NPs, formed by the spontaneous aggregation in water of compounds made by the conjugation of the drug of choice to a self-assembly inducer [1]. These nanostructures, in which the drug moiety is already contained in its building blocks instead of being loaded on inert carriers, present several advantages: (1) A high and precisely tunable drug-loading capacity; (2) a simple adjustment of the physicochemical features of the NPs by optimizing the molecular design; (3) an easy preparation; and (4) an increased biocompatibility as there is no potential carrier-induced cytotoxicity and immunogenicity [2].

For several years we have been interested in using this kind of NPs to improve the properties of both anticancer and neuroprotective drugs [3,4,5,6,7,8,9,10,11,12,13,14]. We designed conjugates able to form NPs that can release the drug in cellular media, [3,8,11,14] hetero-NPs bearing two different drugs [6,9] and fluorescent NPs obtained mixing drug- and fluorophore-based conjugates [4,5]. In our previous works, squalene, [3,4,5,6,11,12,13] 4-(1,2-diphenylbut-1-en-1-yl)aniline, [7,8,14] 20-hydroxyecdisone [9], or betulinic acid [10] was used. The goal of the present work is to identify a molecule that, besides being able to induce the aggregation, would also present some biological activity to further improve the NPs pharmacological properties.

Cannabidiol (CBD, **1**, Figure 1) is a nonpsychoactive phytocannabinoid compound extracted from flowers or leaves of the plants of the *Cannabis* genus and, in particular, from *Cannabis sativa*. It was first isolated between 1930 and 1940, but its chemical structure was only clarified in 1963. The biological effects of this compound were investigated in the following years and included anticonvulsant, anti-inflammatory, anti-anxiety, and anti-cancer activities, along with beneficial effects for the immune system [15,16]. In particular, CBD, alone or in combination with other agents, has been shown to successfully induce cell death, inhibit cell migration and invasion in vitro, decrease tumor size, vascularization, growth, and weight, and increase survival and induce tumor regression in vivo [17,18,19].

Based on the above statements, we considered CBD for its potential dual activity (cytotoxic compound and self-assembly inducer) and to conjugate it to well-known tubulin binder drugs, *N*-desacetyl thiocolchicine (**2**), podophyllotoxin (**3**), and paclitaxel (**4**), through two different linkers, **5a** and **5b** (Figure 1). The synthesis and characterization of the planned conjugates and their ability to form self-assembled NPs are here reported. The ability to induce an antiproliferative effect was assayed on three human tumor cell lines (MSTO-211H, HT-29 and HepG2) and the maintenance of the cell target was assessed by confocal microscopy.

## 2. Results and Discussion

The following Figure 1 shows the retrosynthetic approach for the preparation of the CBD-based conjugates **6–8a,b.**

Said conjugates should be obtained through a condensation reaction between one among the cytotoxic drug-linker intermediates **9–11a,b** and CBD **1**. Compounds **9–11a,b** could derive from the deprotection of carboxy-protected intermediates **12–14a,b**, which could be formed as the result of another condensation reaction between the selected drugs and the mono carboxy-protected esters **15a,b**. Eventually, said compounds could be protected by reacting the linkers **5a,b** with the proper protecting group under esterification conditions. Starting with the description of the conjugates obtained involving *N*-desacetyl thiocolchicine **2** as the drug, the pathway depicting their synthesis is shown below in Figure 2. As the conjugates differ only by the linker (being either sebacic or 4,4′-dithiodibutyric acid), their synthesis was carried out following the same protocol: the initial protection of the dicarboxylic acids was avoided and the direct reaction was tried leading to the **9a,b** intermediates in good enough yields (step a, Figure 2). The latter were coupled with our potential self-assembly inducer **1** under Steglich esterification conditions leading to final products **6a,b** (step b, Figure 2)**.**


In both syntheses, diamides and diesters were the principal generated by-products, even though all the condensation reactions were performed with excess of the moiety presenting the double functionality.

Then, we will consider the synthesis of the two compounds **7a,b**, presenting podophyllotoxin as the drug, as shown in Figure 3.

The first step of the synthesis is the selective protection of one of two carboxylic acids of sebacic acid **5a** and 4,4′-dithiodibutyric acid **5b** (step a, Figure 3). This reaction was needed as it was seen that, in this case, the use of dicarboxylic acids under condensation conditions led to the formation of significant amounts of dimeric side products, losing precious quantities of drugs and complicating the purification of target compounds. We decided to use 2-(trimethylsilyl)ethanol (0.3 eq.) as the protective group in a Steglich esterification. It is to be noted that even using low amounts of esterifying agent to reduce the diesterification that can occur, we obtained both target monoesters **16a,b** and minoritary diester by-products. We then proceeded with the coupling of these monoprotected linkers with the drug podophyllotoxin **3**, using Steglich esterification, obtaining TMSE-protected linker-drug conjugates **17a,b** in good yields (step b, Figure 3). Conjugates **17a,b** were then deprotected with TBAF to obtain the corresponding free carboxylic acids **10a,b** with acceptable yields (step c, Figure 3). The final step of this synthetic pathway is the coupling of CBD, the self-assembly inducer, with the conjugates **10a,b**. The esterification reaction is again performed under Steglich conditions in the presence of EDC·HCl and DMAP, leading to the obtainment of target compounds **7a,b** (step d, Figure 3). For both conjugates the diesterification by-products were not detected, probably due to the restricted rotation of the CBD cyclohexenyl moiety that could shield the second free hydroxy group.

The last conjugates to be synthetized were the two paclitaxel-based derivatives. At first, we followed the strategy used to synthesize conjugates **7a,b**: the TMSE-protected esters **16a,b** were reacted with **4** under Steglich esterification conditions to obtain intermediates **18a,b** quantitatively (step a, Figure 4). We then studied the cleavage of the silylated protecting group to obtain the corresponding carboxylic acids **11a,b** (step b, Figure 4). The deprotection was performed with different experimental conditions. Unfortunately, all the studied conditions led to the degradation of the products. This result is probably generated by the instability of paclitaxel in the presence of TBAF. As the previous synthetic strategy failed, we decided to change the protecting group of the linker into another one that could be removed in different conditions.

We then protected one of the two carboxylic acids of sebacic acid **5a** and of 4,4′-dithiodibutyric acid **5b** as 2,2,2-trichloroethyl esters (TCEs), which are cleavable in reductive conditions, with metallic zinc at mildly acidic pH (step a, Figure 5). We proceeded again with the coupling to the drug, performing once again a Steglich esterification in the presence of **4** with the usual conditions, obtaining the derivatives **19a,b** with high yields (step b, Figure 5). At this point, we tried the reductive cleavage of the previously synthesized conjugates **19a,b** to obtain the corresponding carboxylic acids **11a,b** (step c, Figure 5). For what regards the paclitaxel-sebacic-TCE **19a**, it was dissolved in a 1:1 mixture of glacial acetic acid/methanol at room temperature, and after adding a large excess of zinc dust and waiting four hours, we could successfully isolate the target carboxylic acid **11a** with a 57% yield (step c, Figure 5).

Unfortunately, performing the same reaction on protected ester **19b** did not lead to the same result, as we assisted at the degradation of the starting material with no formation of desired compound **11b**. This may be due to the presence of the disulfide bond, which could be particularly sensitive in the presence of zinc, which shows a great tendency to form Zn-S bonds (soft-soft interactions). However, we were able to complete the synthesis with conjugate **11a**, performing its conjugation with cannabidiol **1** and obtaining target compound **8a** in moderate yield (step d, Figure 5). 

Since all the previous attempts failed when we tried to synthesize the target conjugate **8b**, as the concomitant presence of the paclitaxel and disulfide moiety makes the molecule more sensitive and delicate under various conditions, we decided to change the synthetic strategy. In Figure 6, the second synthetic strategy is reported. Here the order of the condensation steps is reversed, as we first reacted cannabidiol with the protected linker and only then with paclitaxel. We started this different synthetic approach with the monoesterification under Steglich conditions between two equivalents of **1** and the already prepared mono-protected TMSE-carboxylic acid **16b** (step a, Figure 6), trying to limit the diesterification by-product formation. The desired conjugate **20b** was successfully obtained with a 57% yield, as despite the cannabidiol excess, we also obtained 11% of the undesired diesterification by-product. The following step was the deprotection of intermediate **20b** using TBAF to achieve the carboxylic acid **21b** (step b, Figure 6). Having the linker-self-assembly inducer conjugate **21b** in our hands, we could perform the last reaction step to obtain the target compound **8b**, which consisted once again in a Steglich coupling between paclitaxel and the carboxylic acid **21b**. Although the CBD conjugate contains another free hydroxy group, we could obtain the target derivative **7–4c** with a good yield (step c, Figure 6).

To test the self-assembly ability imparted by cannabidiol to the obtained conjugates **6–8a,b**, we proceeded with the preparation and characterization of their nanoparticles (Figure 2).

The six nanosuspensions—one for each **6–8a,b** conjugate—were prepared in accordance with standard solvent evaporation protocols [20]. The NP suspensions **6–8A,B** obtained were then characterized for what regards their bio-physical properties. DLS and Z-potential measurements were carried out on each NP after 10 minutes’ sonication, giving the following results (Table 1).

DLS confirmed the formation of nanoassemblies in aqueous medium. Namely, the low polydispersity index values (PI < 0.2) indicated that each CBD-linker-drug conjugate **6–8a,b** was able to give monodisperse suspensions of NPs, with hydrodynamic diameters (HDs) in the 160–400 nm range. Even though the dimension of some of them is around the higher end of NPs’ definition (500 nm), we expect them to be able to exert their action and be internalized in cells. The zeta potential was negative (<−25 mV) for all the nanoassemblies, suggesting that electrostatic repulsion contributes to the colloidal stability of each suspension.

The antiproliferative effect of the obtained NPs was evaluated on three human tumor cell lines, MSTO-211H (biphasic mesothelioma), HT-29 (colorectal adenocarcinoma), and HepG2 (hepatocellular carcinoma). The cytotoxicity exerted by CBD (**1**), *N*-desacetyl thiocolchicine (**2**), podophyllotoxin (**3**), and paclitaxel (**4**) was also determined in all cell lines. The results are expressed as GI_50_ values, that is the concentration inducing a 50% reduction in cell number with respect to a control culture, and are shown in Table 2.

According to the literature data [21], the self-assembly inducer CBD (**1**) shows an antiproliferative effect on human tumor cell lines with GI_50_ values in the micromolar range, while the well-known antiproliferative agents, *n*-desacetylthiocolchicine (**2**), podophyllotoxin (**3**) and paclitaxel (**4**), provoke a strong cytotoxicity, assessed by GI_50_ values in the nanomolar range.

The treatment with NPs **6A**,**B**, **7A**,**B**, and **8A**,**B** induced a cytotoxicity that appears less pronounced with respect to that of the corresponding free drugs, **2**, **3**, and **4**, respectively, with GI_50_ values ranging from 0.43 μM to 14.5 μM. Because the cell effect of NPs depends on the release of the cytotoxic agent, it is possible that the hydrolysis of the linker and the kinetics of such release account for a lower intracellular content with respect to the incubation with free drugs, leading to NPs that are not so effective as pure drugs. Moreover, the antiproliferative activity exerted by the NPs depends on the cell line, with MSTO211-H generally the most sensitive and in particular, the lowest GI_50_ value, 0.43 μM, was obtained by treating this cell line with **8B**. This result appears in agreement with the literature data demonstrating that the treatment with paclitaxel-loaded NPs prolonged the survival of a murine model of malignant pleural mesothelioma, obtained by intrathoracic injection of MSTO-211H cells [22]. Overall, these results suggest that the poor clinical response of mesothelioma to paclitaxel could be attributable to an ineffective delivery kinetics of the systemic administration rather than to the mechanism of drug itself, and allow to consider the use of NPs extremely interesting for the development of effective antitumor therapy. The antiproliferative effect appears to depend also on the type of linker. In this connection, it is to note that NPs prepared with conjugates carrying 4,4′-dithiodibutyric acid as the linker and then characterized by the presence of a disulfide bond (**6B**, **7B** and **8B**) are remarkably more effective in inducing cell death with respect to the corresponding NPs where the linker of the conjugates is the sebacic acid (**6A**, **7A** and **8A**). The more pronounced difference in activity emerges by comparing **6A** vs. **6B** and **8A** vs. **8B**, and indeed, in these NP pairs the presence of the disulfide bond induces a decrease in GI_50_ values of about 11 and 22 times, respectively. 

Furthermore, the treatment of MeT-5A, human mesothelial cells, with paclitaxel allowed to obtain a GI_50_ value of 3.2 ± 0.6 nM, similar to that of human mesothelioma MSTO-211H (3.5 ± 0.2 nM, Table 2), confirming the absence of tumor selectivity by the drug. Otherwise, the incubation of MeT-5A with **8B** showed a GI_50_ value of 2.1 ± 0.5 µM, that is about four times higher than that obtained for mesothelioma tumor cells (0.43 ± 0.01 µM, Table 2), indicating a lower effect in normal cells. These data are in agreement with the observation that intracellular levels of reduced glutathione are upregulated in a number of human cancers [23] and support the rationale of the synthetic approach, in accordance with the ability of the disulfide-containing bivalent conjugates to release the active drug inside cells, as already reported [3,11].

The interesting cell effect of **8B** suggested us to investigate its possible mechanism of action, taking into account that the major intracellular effect of paclitaxel is the kinetic suppression of microtubule dynamics and then their stabilization [24]. For this purpose, confocal microscopy experiments were performed in the most sensitive MSTO-211H cells treated with **8B**. Figure 3 shows the results obtained by incubating cells in standard conditions (A,B), in the presence of free paclitaxel (C,D) or **8B** (E,F) for 4 h at 1 μM and 100 μM, respectively, that is a difference in concentration comparable to that observed between the GI_50_ values (see Table 2).

As shown by confocal microscope imaging, the cells in control condition (Figure 3A,B) exhibited a normal microtubule array with filamentous microtubules distributed in the cytoplasm. As expected, treatment with paclitaxel (Figure 3C,D), induced the formation of a highly organized network of microtubules with the formation of long microtubule bundles, mainly surrounding the nucleus. In the presence of **8B**, a partial displacement of the microtubule network present in non-treated cells is observed, with the onset of a concurrent microtubule aggregation (Figure 3E,F).

It is well-known that cell death induced by paclitaxel occurs through multiple mechanisms that depend on cell type, concentrations, and cell cycle stage [25,26,27].

Starting from these considerations and based on the interesting cytotoxic effect of **8B**, we investigated the cell death mediated by the NP, in comparison with paclitaxel, to assess the mechanism of cell death. In detail, the most sensitive MSTO-211H were incubated with **8B** or paclitaxel for 24 h at a concentration about three times higher with respect to the GI_50_ value, stained with Annexin V-FITC/propidium iodide and analyzed by flow cytometry. Annexin V stains apoptotic cells by binding to phosphatidylserine, a marker of apoptosis, while propidium iodide stains late apoptotic and necrotic cells because it is internalized only in cells undergoing the loss of plasma and nuclear membrane integrity. Figure 4 shows the dot plots of a representative experiment (A) and the percentages of viable, apoptotic, and necrotic cells (B).

The treatment with paclitaxel provokes a clear decrease in cell viability, from about 85% in control condition to about 60%, accompanied by a significant increase in both apoptotic and necrotic cells, more than 20% and 15%, respectively. A similar behavior is also observed in cells incubated in the presence of **8B**. In this experimental condition, the decrease in viability is pronounced, with about 50% of cell death, which occurs through both apoptosis (35%) and necrosis (10%) pathway.

These results confirm the involvement of paclitaxel in cytotoxicity and cell death mechanism mediated by **8B**, supporting the rationale of the approach and confirming the ability of the NP to address the cytotoxic drug inside the cell, allowing the related effect.

## 3. Materials and Methods

All reactions were carried out in oven-dried glassware and dry solvents under nitrogen atmosphere. Unless otherwise stated, all solvent and reagents were purchased from Sigma-Aldrich (Milan, Italy), Fluorochem (Hadfield, UK), or TCI (Zwijndrecht, Belgium) and used without further purification. CBD was extracted through dynamic maceration (DM) of *Cannabis sativa* inflorescences using EtOH, following a published protocol [28]. HPLC analysis was performed to assess its purity (>95%) on an ASCENTIS RP-C_18_ column (5 μm × 4.6 × 150 mm). The pressure was set at about 101 bar, and the temperature was maintained at 40 °C, with a constant flow rate of 0.95 mL/min. UV spectra were recorded at 228 nm using a gradient elution method. The mobile phase consisted of a mixture of A (0.1% *v*/*v* HCOOH in H_2_O) and B (0.1% *v*/*v* HCOOH in MeCN). The gradient elution program was adapted to a 30 min duration to obtain RRT 1.00 for CBD, following a published protocol [29]. Thin layer chromatography (TLC) was performed on Merck precoated 60F254 plates. Reactions were monitored by TLC on silica gel, with detection by UV light (254 nm) or by staining with p-anisaldehyde or potassium permanganate solutions with heating. Purification of intermediates and final products was mostly carried out by flash chromatography using as stationary phase high purity grade silica gel (Merck Grade, pore size 60 Å, 230–400 mesh particle size, Sigma Aldrich). Alternatively, purification was performed by a Biotage^®^ system in normal phase using Biotage^®^ Sfär Silica D cartridges (10/25 g). Intermediates and final products were structurally characterized by ^1^H NMR and 13C NMR spectroscopy at 400/500 MHz, using a Bruker AC 400/500 spectrometer. Chemical shifts (δ) for proton and carbon resonances are quoted in parts per million (ppm) relative to tetramethylsilane (TMS), which was used as an internal standard. ^1^H and ^13^C spectra of all synthetized compounds can be found in Appendix A. HRMS spectra were recorded using electrospray ionization (ESI) technique on a FT-ICR APEXII (Bruker Daltonics, Bremen, Germany). Specific rotations were measured with a P-1030-Jasco polarimeter with 10 cm optical path cells and 1 mL capacity (Na lamp, λ = 589 nm).

### 3.1. Synthetic Procedures

#### General Procedure for Steglich Coupling

To a stirred solution of the corresponding carboxylic acid (1.0–1.2 eq) in dry dichloromethane (CH_2_Cl_2_) (0.02–0.05 M), *N*-(3-dimethylaminopropyl)-*N′*-ethylcarbodiimide hydrochloride (EDC∙HCl) (1.5 eq), and *N*,*N*-dimethyl-4-aminopyridine (DMAP) (0.5 eq) are added under nitrogen atmosphere and at r.t. The reaction is stirred for 15 min, and the corresponding alcohol (1.0–1.2 eq) is added. The reaction mixture is then left stirring overnight at r.t. After reaction completion (TLC monitoring), 1M HCl solution is added, and the mixture is extracted with CH_2_Cl_2_. The collected organic phases are then dried over anhydrous Na_2_SO_4_, filtered, and concentrated under reduced pressure. The crude reaction mixtures are eventually purified by flash column chromatography or Biotage^®^, when needed.


**4-((4-oxo-4-(2-(trimethylsilyl)ethoxy)butyl)disulfaneyl)butanoic acid (16b)**


Following the general procedure for the Steglich coupling, to a solution of 4,4′-dithiodibutyric acid **5b** (1000 mg, 4.20 mmol), EDC∙HCl (472 mg, 2.46 mmol), DMAP (75 mg, 0.62 mmol) in dry CH_2_Cl_2_ (25 mL, 0.1 M) and pyridine (2.5 mL), trimethylsilyl ethanol (0.26 mL, 1.84 mmol) is added to obtain 503 mg of product **16b** with 81% yield. Reaction is monitored by TLC (7:3 *n*-hex/EtOAc + 1% HCOOH) and purified by flash chromatography (8:2 *n*-hex/EtOAc + 1% HCOOH).

**^1^H NMR (400 MHz, CDCl_3_)** δ 4.17 (m, 2H), 2.73 (m, 4H), 2.52 (t, *J* = 7.3 Hz, 2H), 2.42 (t, *J* = 7.3 Hz, 2H), 2.03 (m, 4H), 0.98 (m, 2H), 0.04 (s, 9H).

**HRMS (ESI^+^):** *m/z* [M + Na]^+^ calcd. for C_13_H_26_O_4_S_2_SiNa, 361.0939; found, 361.0940.

Spectroscopic data are consistent with those described in the literature [9].


**10-oxo-10-(2-(trimethylsilyl)ethoxy)decanoic acid (16a)**


Following the general procedure for the Steglich coupling, to a solution of sebacic acid **5a** (1000 mg, 4.94 mmol), EDC∙HCl (559 mg, 2.92 mmol), DMAP (89 mg, 0.73 mmol) in dry CH_2_Cl_2_ (25 mL, 0.1 M) and pyridine (2.5 mL), trimethylsilyl ethanol (0.31 mL, 2.18 mmol) were added to obtain 560 mg of product **16a** with 85% yield. Reaction is monitored by TLC (7:3 *n*-hex/EtOAc + 1% HCOOH) and purified by flash chromatography (8:2 *n*-hex/EtOAc + 1% HCOOH).

**^1^H NMR (400 MHz, CDCl_3_)** δ 4.20–4.11 (m, 2H), 2.34 (t, *J* = 7.5 Hz, 2H), 2.27 (t, *J* = 7.5 Hz, 2H), 1.69–1.55 (m, 4H), 1.37–1.25 (m, 8H), 1.03–0.93 (m, 2H), 0.04 (s, 9H).

**^13^C NMR (100 MHz, CDCl_3_)** δ 180.2, 174.2, 62.5, 34.6, 34.2, 29.2, 29.1, 29.1, 25.0, 24.7, 17.4, −1.4.

**HRMS (ESI^+^):** *m/z* [M + Na]^+^ calcd. for C_15_H_30_O_4_SiNa, 325.1811; found, 325.1815.


**4-((4-oxo-4-(2,2,2-trichloroethoxy)butyl)disulfaneyl)butanoic acid (18b)**


Following the general procedure for the Steglich coupling, to a solution of 4,4′-dithiodibutyric acid **5b** (1.56 g, 6.54 mmol), EDC∙HCl (751 mg, 3.92 mmol), DMAP (478 mg, 3.91 mmol) in dry CH_2_Cl_2_ (52 mL, 0,05 M), 2,2,2-trichloro ethanol (0.25 mL, 2.61 mmol) is added to obtain 567 mg of product **18b** with 59% yield. Reaction is monitored by TLC (7:3 *n*-hex/EtOAc + 1% HCOOH) and purified by flash chromatography (8:2 *n*-hex/EtOAc + 1% HCOOH).

**^1^H NMR (400 MHz, CDCl_3_)** δ 4.75 (s, 2H), 2.74 (q, *J* = 7.2 Hz, 4H), 2.61 (t, *J* = 7.2 Hz, 2H), 2.50 (t, *J* = 7.2 Hz, 2H), 2.16–1.98 (m, 4H).

**^13^C NMR (100 MHz, CDCl_3_)** δ 179.4, 171.4, 95.0, 74.0, 37.6, 37.6, 32.4, 32.3, 24.0, 23.9.

**HRMS (ESI^+^):** *m/z* [M + Na]^+^ calcd. for C_10_H_15_Cl_3_O_4_S_2_Na, 390.9375; found, 390.9377.


**10-oxo-10-(2,2,2-trichloroethoxy)decanoic acid (18a)**


Following the general procedure for the Steglich coupling, to a solution of sebacic acid **5a** (1.32 g, 6.53 mmol), EDC∙HCl (750 mg, 3.91 mmol), DMAP (480 mg, 3.93 mmol) in dry CH_2_Cl_2_ (52 mL, 0.05 M), 2,2,2-trichloro ethanol (0.25 mL, 2.61 mmol) is added to obtain 472 mg of product **18a** with 52% yield. Reaction is monitored by TLC (7:3 *n*-hex/EtOAc + 1% HCOOH) and purified by flash chromatography (8:2 *n*-hex/EtOAc + 1% HCOOH).

**^1^H NMR (400 MHz, CDCl_3_)** δ 4.74 (s, 2H), 2.46 (t, *J* = 7.5 Hz, 2H), 2.35 (t, *J* = 7.5 Hz, 2H), 1.66 (m, 4H), 1.40–1.26 (m, 8H).

**^13^C NMR (100 MHz, CDCl_3_)** δ 180.4, 172.2, 95.2, 73.9, 34.1, 34.0, 29.1, 29.0, 29.0, 29.0, 24.8, 24.7.

**HRMS (ESI^+^):** *m/z* [M + Na]^+^ calcd. for C_12_H_19_Cl_3_O_4_Na, 355.0247; found, 355.0248


**(*S*)-10-oxo-10-((1,2,3-trimethoxy-10-(methylthio)-9-oxo-5,6,7,9-tetrahydrobenzo[*a*]heptalen-7-yl)amino)decanoic acid (9a)**


Following the general procedure for the Steglich coupling, to a solution of carboxylic acid **5a** (135 mg, 0.67 mmol), EDC∙HCl (56 mg, 0.29 mmol), DMAP (4 mg, 0.03 mmol), TEA (0.30 mL, 2.14 mmol) in dry CH_2_Cl_2_ (3.0 mL, 0.09 M), **2** (100 mg, 0.27 mmol) is added to obtain 93 mg of product **9a** with 62% yield. Reaction is monitored by TLC (CH_2_Cl_2_/MeOH 95:5) and purified by flash chromatography (CH_2_Cl_2_/MeOH 95:5 eluent mixture).

**^1^H NMR (400 MHz, CDCl_3_)** δ 7.60 (s, 1H), 7.44 (s, 1H), 7.34 (d, *J* = 10.5 Hz, 1H), 7.11 (d, *J* = 10.5 Hz, 1H), 6.53 (s, 1H), 4.71 (dt, *J* = 12.8, 6.7 Hz, 1H), 3.93 (s, 3H), 3.89 (s, 3H), 3.65 (s, 3H), 2.51 (dd, *J* = 13.2, 5.9 Hz, 1H), 2.43 (s, 3H), 2.41–2.25 (m, 3H), 2.26–2.17 (m, 3H), 1.95–1.84 (m, 1H), 1.67–1.49 (m, 4H), 1.34–1.21 (m, 8H).

**HRMS (ESI^+^):** *m/z* [M + Na]^+^ calcd. for C_30_H_39_NO_7_SNa, 580.2345; found, 580.2348.

Spectroscopic data are consistent to the ones reported in the literature [30].


**(1’*R*,2’*R*)-6-hydroxy-5’-methyl-4-pentyl-2’-(prop-1-en-2-yl)-1’,2’,3’,4’-tetrahydro-[1,1’-biphenyl]-2-yl 10-oxo-10-(((*S*)-1,2,3-trimethoxy-10-(methylthio)-9-oxo-5,6,7,9-tetrahydrobenzo[*a*]heptalen-7-yl)amino)decanoate (6a)**


Following the general procedure for the Steglich coupling, to a solution of carboxylic acid **9a** (50 mg, 0.09 mmol), EDC∙HCl (19 mg, 0.98 mmol), DMAP (1 mg, 0.01 mmol), TEA (0.01 mL, 0.98 mmol) in dry CH_2_Cl_2_ (2 mL, 0.04 M), **1** (28 mg, 0.09 mmol) are added to obtain 36 mg of product **6a** with 45% yield. Reaction is monitored by TLC (*n*-hex/EtOAc 1:1) and purified by flash chromatography (*n*-hex/EtOAc 1:1).

**^1^H NMR (400 MHz, CDCl_3_)** δ 7.36–7.27 (m, 2H), 7.08 (d, *J* = 10.5 Hz, 1H), 6.53 (s, 2H), 6.45 (d, *J* = 7.4 Hz, 1H), 6.37 (s, 1H), 5.95 (bs, 1H), 5.51 (bs, 1H), 4.67 (dt, *J* = 13.2, 6.8 Hz, 1H), 4.62–4.57 (m, 1H), 4.44 (bs, 1H), 3.94 (s, 3H), 3.90 (s, 3H), 3.66 (s, 3H), 3.49 (bs, 1H), 2.59–2.32 (m, 9H), 2.31–2.13 (m, 4H), 2.11–1.64 (m, 12H), 1.65–1.50 (m, 3H), 1.39–1.22 (m, 14H), 0.88 (t, *J* = 7.7 Hz, 3H).

**^13^C NMR (100 MHz, CDCl_3_)** δ 182.2, 173.0, 172.2, 158.4, 153.8, 152.0, 151.3, 141.8, 138.8, 135.8, 135.1, 134.5, 129.9, 128.6, 127.8, 127.0, 125.8, 123.5, 114.5, 111.4, 107.5, 61.8, 61.5, 56.2, 52.0, 45.7, 37.9, 37.0, 36.5, 35.5, 34.4, 31.6, 30.6, 30.3, 30.1, 29.8, 29.4, 29.2, 28.1, 27.0, 25.6, 24.9, 23.7, 22.6, 19.9, 15.3, 14.1.

**HRMS (ESI^+^):***m/z* [M + Na]^+^ calcd. for C_51_H_67_NO_8_SNa, 876.4485; found, 876.4491.

${\left[\alpha \right]}_{D}^{25}$**:** -72.8 (*c* 1 in CHCl_3_).


**(*S*)-4-((4-oxo-4-((1,2,3-trimethoxy-10-(methylthio)-9-oxo-5,6,7,9-tetrahydrobenzo[*a*] heptalen-7-yl)amino)butyl)disulfaneyl)butanoic acid (9b)**


Following the general procedure for the Steglich coupling, to a solution of carboxylic acid **5b** (238 mg, 1.0 mmol), EDC∙HCl (84 mg, 0.44 mmol), DMAP (5 mg, 0.04 mmol), TEA (0.45 mL, 3.2 mmol) in dry CH_2_Cl_2_ (4.5 mL, 0.01 M), **2** (150 mg, 0.40 mmol) is added to obtain 117 mg of product **9b** with 49% yield. Reaction is monitored by TLC (CH_2_Cl_2_/MeOH 98:2 + 1% HCOOH) and purified by Biotage^®^ (gradient with *n*-hex/EtOAc eluent mixture).

**^1^H NMR (400 MHz, CDCl_3_)** δ 8.27 (s, 1H), 7.56 (d, *J* = 10.7 Hz, 1H), 7.33 (d, *J* = 10.7 Hz, 1H), 6.56 (s, 1H), 4.85–4.72 (m, 1H), 3.97–3.89 (m, 6H), 3.76–3.59 (m, 3H), 2.88–2.67 (m, 5H), 2.63–2.41 (m, 9H), 2.43–2.26 (m, 2H), 2.13–1.88 (m, 4H).

**HRMS (ESI^+^):** *m/z* [M + Na]^+^ calcd. for C_28_H_35_NO_7_S_3_Na, 616.1473; found, 616.1478.

Spectroscopic data are consistent to the ones reported in literature [31].


**(1’*R*,2’*R*)-6-hydroxy-5’-methyl-4-pentyl-2’-(prop-1-en-2-yl)-1’,2’,3’,4’-tetrahydro-[1,1’-biphenyl]-2-yl 4-((4-oxo-4-(((*S*)-1,2,3-trimethoxy-10-(methylthio)-9-oxo-5,6,7,9-tetrahydrobenzo[*a*]heptalen-7 yl)amino)butyl)disulfaneyl)butanoate (6b)**


Following the general procedure for the Steglich coupling, to a solution of carboxylic acid **9b** (70 mg, 0.18 mmol), EDC∙HCl (38 mg, 0.20 mmol), DMAP (3 mg, 0.02 mmol), TEA (0.2 mL, 1.44 mmol) in dry CH_2_Cl_2_ (4 mL, 0.01 M), **1** (57 mg, 0.18 mmol) is added to obtain 32 mg of product **6b** with 20% yield. Reaction is monitored by TLC (*n*-hex/EtOAc 3:7) and purified by flash chromatography (*n*-hex/EtOAc 3:7 + 1% HCOOH).

**^1^H NMR (400 MHz, CDCl_3_)** δ 7.37–7.27 (m, 2H), 7.07 (d, *J* = 10.5 Hz, 1H), 6.99–6.83 (m, 1H), 6.76–6.65 (m, 1H), 6.53 (bs, 2H), 6.37 (s, 1H), 5.97 (bs, 1H), 5.47 (bs, 1H), 4.64 (dt, *J* = 11.9, 6.6 Hz, 1H), 4.61–4.56 (m, 1H), 4.44 (s, 1H), 3.93 (s, 3H), 3.89 (s, 3H), 3.66 (s, 3H), 2.83–2.72 (m, 3H), 2.72–2.60 (m, 4H), 2.55–2.41 (m, 7H), 2.42–2.29 (m, 2H), 2.27–2.05 (m, 4H), 1.99 (q, *J* = 7.3 Hz, 2H), 1.85–1.68 (m, 5H), 1.62–1.47 (m, 3H), 1.36–1.18 (m, 8H), 0.97–0.77 (m, 3H).

**^13^C NMR (100 MHz, CDCl_3_)** δ 182.4, 171.8, 171.7, 158.5, 155.7, 153.8, 151.4, 142.9, 141.8, 138.5, 134.9, 134.5, 128.6, 126.8, 125.8, 124.5, 114.7, 111.5, 107.6, 61.8, 61.5, 56.2, 52.1, 45.8, 38.4, 38.0, 37.6, 36.9, 35.5, 35.3, 34.6, 32.5, 31.6, 30.6, 30.5, 30.3, 30.1, 29.8, 24.7, 24.7, 24.3, 23.8, 22.6, 20.0, 15.3, 14.2.

**HRMS (ESI^+^):***m/z* [M + Na]^+^ calcd. for C_49_H_63_NO_8_S_3_Na, 912.3613; found, 912.3619.

${\left[\alpha \right]}_{D}^{25}$**:** -117.1 (*c* 0.96 in CHCl_3_).


**(5*R*,5a*R*,8a*R*,9*R*)-8-oxo-9-(3,4,5-trimethoxyphenyl)-5,5a,6,8,8a,9-hexahydrofuro[3’,4’:6,7] naphtho[2,3-*d*][1,3]**
**dioxol-5-yl 4-((4-oxo-4-(2-(trimethylsilyl)ethoxy)butyl)disulfaneyl) butanoate (17b)**


Following the general procedure for the Steglich coupling, to a solution of carboxylic acid **16b** (74.0 mg, 0.22 mmol), EDC∙HCl (41.7 mg, 0.22 mmol), DMAP (11.0 mg, 0.09 mmol) in dry CH_2_Cl_2_ (9 mL, 0.02 M), **3** (75 mg, 0.18 mmol) is added to obtain 1 mg of product **17b** quantitatively. Reaction is monitored by TLC (4:6 *n*-hex/EtOAc) and purified by Biotage^®^ (gradient with *n*-hex/EtOAc eluent mixture).

**^1^H NMR (400 MHz, CDCl_3_)** δ 6.74 (s, 1H), 6.51 (s, 1H), 6.36 (s, 2H), 5.95 (d, *J* = 5.4 Hz, 2H), 5.86 (d, *J* = 9.0 Hz, 1H), 4.57 (d, *J* = 3.9 Hz, 1H), 4.39–4.30 (m, 1H), 4.22–4.09 (m, 3H), 3.77 (s, 3H), 3.73 (s, 6H), 2.90 (dd, *J* = 14.5, 4.5 Hz, 1H), 2.87–2.75 (m, 1H), 2.77–2.65 (m, 4H), 2.65–2.45 (m, 2H), 2.38 (t, *J* = 7.4 Hz, 2H), 2.12–1.93 (m, 4H), 1.00–0.91 (m, 2H), 0.01 (s, 9H).

**^13^C NMR (100 MHz, CDCl_3_)** δ 173.6, 173.3, 172.9, 152.5 (2C), 148.1, 147.5, 137.0, 134.8, 132.3, 128.2, 109.6, 108.0 (2C), 106.9, 101.6, 73.6, 71.3, 62.6, 60.6, 56.1 (2C), 45.4, 43.6, 38.6, 37.7, 37.6, 32.7, 32.5, 24.2, 24.0, 17.3, −1.5 (3C).

**HRMS (ESI^+^):***m/z* [M + Na]^+^ calcd. for C_35_H_46_O_11_S_2_SiNa, 757.2148; found, 757.2150.


**4-((4-oxo-4-(((5*R*,5a*R*,8a*R*,9*R*)-8-oxo-9-(3,4,5-trimethoxyphenyl)-5,5a,6,8,8a,9-hexahydrofuro[3’,4’:6,7]naphtho[2,3-*d*][1,3]dioxol-5-yl)oxy)butyl)disulfaneyl)butanoic acid (10b)**


To a well-stirred solution of compound **17b** (141 mg, 0.192 mmol) in dry THF (3.9 mL, 0.05M), 1M TBAF in THF (1.92 mL, 1.92 mmol) is added at 0 °C under nitrogen atmosphere. The reaction is stirred at r.t. and monitored by TLC (*n*-hex/EtOAc 1:1 + 1% HCOOH). After 1 h, the reaction is quenched with saturated NH_4_Cl aqueous solution and extracted with abundant CH_2_Cl_2_. The organic phase is then dried over anhydrous Na_2_SO_4_, filtered, and evaporated under reduced pressure. The crude mixture is then purified by Biotage^®^ (gradient with *n*-hex/EtOAc + 1% HCOOH eluent mixture) to obtain 59 mg of the desired product **10b** (48% yield).

**^1^H NMR (400 MHz, CDCl_3_)** δ 6.74 (d, *J* = 5.0 Hz, 1H), 6.54 (s, 1H), 6.38 (s, 2H), 5.99–5.92 (m, 2H), 5.73 (d, *J* = 4.9 Hz, 1H), 4.47–4.33 (m, 2H), 4.28 (dd, *J* = 9.6, 3.0 Hz, 1H), 3.82 (s, 3H), 3.80 (s, 6H), 3.29 (dd, *J* = 9.2, 3.6 Hz, 1H), 3.03–2.92 (m, 1H), 2.79–2.62 (m, 4H), 2.52–2.45 (m, 2H), 2.43–2.25 (m, 2H), 2.08–1.90 (m, 4H).

**^13^C NMR (100 MHz, CDCl_3_)** δ 178.5, 177.5, 172.7, 153.3 (2C), 148.5, 147.3, 139.0, 136.9, 131.3, 126.2, 109.9, 108.4, 105.5 (2C), 101.5, 72.6, 70.9, 61.0, 56.3 (2C), 45.5, 44.3, 39.8, 37.8, 37.6, 32.5, 32.3, 24.0, 24.0.

**HRMS (ESI^+^):** *m/z* [M + Na]^+^ calcd. for C_30_H_34_O_11_S_2_Na, 657.1440; found, 657.1443.


**(1’*R*,2’*R*)-6-hydroxy-5’-methyl-4-pentyl-2’-(prop-1-en-2-yl)-1’,2’,3’,4’-tetrahydro-[1,1’-biphenyl]-2-yl 4-((4-oxo-4-(((5*R*,5a*R*,8a*R*,9*R*)-8-oxo-9-(3,4,5-trimethoxyphenyl)-5,5a,6, 8,8a,9-hexahydrofuro[3’,4’:6,7]naphtho[2,3-*d*][1,3]dioxol-5-yl)oxy)butyl)disulfaneyl)butanoate (7b)**


Following the general procedure for the Steglich coupling, to a solution of carboxylic acid **10b** (59 mg, 0.093 mmol), EDC∙HCl (22 mg, 0.112 mmol), DMAP (6 mg, 0.047 mmol) in dry CH_2_Cl_2_ (2 mL, 0.05 M), **1** (44 mg, 0.140 mmol) is added to obtain 69 mg of product **7b** with 80% yield. Reaction is monitored by TLC (1:1 *n*-hex/EtOAc) and purified by Biotage^®^ (gradient with *n*-hex/EtOAc eluent mixture).

**^1^H NMR (400 MHz, CDCl_3_)** δ 6.73 (s, 1), 6.54 (s, 2H), 6.40–6.37 (m, 3H), 5.95 (dd, *J* = 10.4, 1.4 Hz, 2H), 5.74 (d, *J* = 5.0 Hz, 1H), 5.52 (s, 1H), 4.63–4.57 (m, 1H), 4.48–4.35 (m, 3H), 4.28 (dd, *J* = 9.6, 2.9 Hz, 1H), 3.83 (s, 3H), 3.80 (s, 6H), 3.47 (s, 1H), 3.28 (dd, *J* = 9.2, 3.6 Hz, 1H), 3.02–2.92 (m, 1H), 2.83–2.72 (m, 2H), 2.68 (t, *J* = 7.0 Hz, 2H), 2.64 (s, 2H), 2.58–2.44 (m, 3H), 2.44–2.27 (m, 2H), 2.28–2.16 (m, 1H), 2.17–2.05 (m, 3H), 1.98 (p, *J* = 7.1 Hz, 2H), 1.87–1.69 (m, 5H), 1.63–1.50 (m, 5H), 1.36–1.24 (m, 4H), 0.87 (t, *J* = 6.8 Hz, 3H).

**^13^C NMR (100 MHz, CDCl_3_)** δ 177.3, 172.7, 171.4, 153.4 (2C), 148.6, 147.4, 143.0, 142.2, 139.0, 133.0, 131.4, 126.3, 123.3, 114.8, 114.1, 111.5, 111.2, 110.0, 108.3, 105.6 (2C), 101.6, 72.6, 70.9, 61.0, 56.4 (2C), 53.6, 45.7, 45.6, 44.3, 39.9, 38.1, 37.6 (2C), 35.5, 32.5 (2C), 31.6, 30.6, 30.3, 29.8, 28.0, 24.1, 24.0, 23.7, 22.6, 14.1.

**HRMS (ESI^+^):** *m/z* [M + Na]^+^ calcd. for C_51_H_62_O_12_S_2_Na, 953.3580; found, 953.3582.

${\left[\alpha \right]}_{D}^{25}$**:** -39.5 (*c* 0.32 in CHCl_3_).


**1-((5*R*,5a*R*,8a*R*,9*R*)-8-oxo-9-(3,4,5-trimethoxyphenyl)-5,5a,6,8,8a,9-hexahydrofuro [3’,4’:6,7]naphtho[2,3-*d*][1,3]dioxol**-5-yl) 10-(2-(trimethylsilyl)ethyl) decanedioate (17a)****


Following the general procedure for the Steglich coupling, to a solution of carboxylic acid **16a** (67 mg, 0.22 mmol), EDC∙HCl (42 mg, 0.22 mmol), DMAP (11 mg, 0.09 mmol) in dry CH_2_Cl_2_ (9 mL, 0.02 M), **3** (75 mg, 0.18 mmol) is added to obtain 91 mg of product **17a** with 72% yield. Reaction is monitored by TLC (4:6 *n*-hex/EtOAc) and purified by Biotage^®^ (gradient with *n*-hex/EtOAc eluent mixture).

**^1^H NMR (400 MHz, CDCl_3_)** δ 6.74 (s, 1H), 6.53 (s, 1H), 6.39 (s, 2H5.98 (dd, *J* = 6.5, 1.3 Hz, 2H), 5.88 (d, *J* = 9.1 Hz, 1H), 4.60 (d, *J* = 4.4 Hz, 1H), 4.36 (dd, *J* = 9.3, 7.0 Hz, 1H), 4.24–4.11 (m, 3H), 3.81 (s, 3H), 3.75 (s, 6H), 2.92 (dd, *J* = 14.5, 4.4 Hz, 1H), 2.88–2.74 (m, 1H), 2.49–2.33 (m, 2H), 2.27 (t, *J* = 7.5 Hz, 2H), 1.73–1.54 (m, 4H), 1.36–1.26 (m, 8H), 1.03–0.92 (m, 2H), 0.03 (s, 9H).

**^13^C NMR (100 MHz, CDCl_3_)** δ 174.2, 174.0, 173.7, 152.7 (2C), 148.1, 147.6, 137.2, 134.9, 132.4, 128.5, 109.7, 108.1 (2C), 107.0, 101.6, 73.4, 71.4, 62.4, 60.8, 56.2 (2C), 45.6, 43.8, 38.8, 34.5, 34.4, 29.1, 29.1, 29.1, 29.1, 25.0, 24.9, 17.3, −1.5 (3C).

**HRMS (ESI^+^):***m/z* [M + Na]^+^ calcd. for C_37_H_50_O_11_SiNa, 721.3020; found, 721.3023.


**10-oxo-10-(((5*R*,5a*R*,8a*R*,9*R*)-8-oxo-9-(3,4,5-trimethoxyphenyl)-5,5a,6,8,8a,9-hexahydrofuro[3’,4’:6,7]naphtho[2,3-*d*][1,3]dioxol-5-yl)oxy)decanoic acid (10a)**


To a well-stirred solution of compound **17a** (139 mg, 0.199 mmol) in dry THF (4.0 mL, 0.05M), 1M TBAF in THF (2.00 mL, 2.00 mmol) is added at 0 °C under nitrogen atmosphere. The reaction is stirred at r.t. and monitored by TLC (*n*-hex/EtOAc 1:1 + 1% HCOOH). After 1 h, the reaction is quenched with saturated NH_4_Cl aqueous solution and extracted with abundant CH_2_Cl_2_. The organic phase is then dried over anhydrous Na_2_SO_4_, filtered, and evaporated under reduced pressure. The crude mixture is then purified by Biotage^®^ (gradient with *n*-hex/EtOAc + 1% HCOOH eluent mixture) to obtain 57 mg of the desired product **10a** (48% yield).

**^1^H NMR (400 MHz, CDCl_3_)** δ 6.71 (s, 1H), 6.52 (s, 1H), 6.39 (s, 2H), 5.94 (dd, *J* = 10.8, 1.4 Hz, 2H), 5.72 (d, *J* = 5.0 Hz, 1H), 4.41 (dd, *J* = 9.6, 6.8 Hz, 1H), 4.36 (d, *J* = 3.7 Hz, 1H), 4.29 (dd, *J* = 9.7, 2.9 Hz, 1H), 3.81 (s, 2H), 3.78 (s, 6H), 3.26 (dd, *J* = 9.2, 3.7 Hz, 1H), 2.99–2.88 (m, 1H), 2.33 (t, *J* = 7.5 Hz, 2H), 2.21 (td, *J* = 7.5, 4.6 Hz, 2H), 1.66–1.49 (m, 4H), 1.35–1.22 (m, 9H).

**^13^C NMR (100 MHz, CDCl_3_)** δ 179.6, 177.4, 173.5, 153.3 (2C), 148.4, 147.3, 138.9, 136.9, 131.3, 126.4, 109.8, 108.2, 105.5 (2C), 101.5, 72.3, 71.0, 60.9, 56.2 (2C), 45.6, 44.2, 39.9, 34.3, 34.0, 29.0 (2C), 29.0, 29.0, 24.8, 24.7.

**HRMS (ESI^+^):** *m/z* [M + Na]^+^ calcd. for C_32_H_38_O_11_Na, 621.2312; found, 621.2313.


**1-((1’*R*,2’*R*)-6-hydroxy-5’-methyl-4-pentyl-2’-(prop-1-en-2-yl)-1’,2’,3’,4’-tetrahydro-[1,1’-biphenyl]-2-yl) 10-((5*R*,5a*R*,8a*R*,9*R*)-8-oxo-9-(3,4,5-trimethoxyphenyl)-5,5a,6,8,8a,9-hexahydrofuro[3’,4’:6,7] naphtho[2,3-*d*][1,3]dioxol-5-yl) decanedioate (7a)**


Following the general procedure for the Steglich coupling, to a solution of carboxylic acid **10a** (59 mg, 0.093 mmol), EDC∙HCl (22 mg, 0.112 mmol), DMAP (6 mg, 0.047 mmol) in dry CH_2_Cl_2_ (2 mL, 0.05 M), **1** (44 mg, 0.140 mmol) is added to obtain 69 mg of product **7a** with 80% yield. Reaction is monitored by TLC (1:1 *n*-hex/EtOAc) and purified by Biotage^®^ (gradient with *n*-hex/EtOAc eluent mixture).

**^1^H NMR (400 MHz, CDCl_3_)** δ 6.72 (s, 1H), 6.56–6.51 (m, 2H), 6.40 (s, 2H), 6.37 (d, *J* = 1.8 Hz, 1H), 5.95 (dd, *J* = 11.0, 1.4 Hz, 2H), 5.74 (d, *J* = 5.1 Hz, 1H), 5.53 (s, 1H), 4.62–4.57 (m, 1H), 4.47–4.35 (m, 3H), 4.31 (dd, *J* = 9.6, 2.8 Hz, 1H), 3.83 (s, 3H), 3.80 (s, 6H), 3.47 (s, 1H), 3.26 (dd, *J* = 9.2, 3.7 Hz, 1H), 2.99–2.89 (m, 1H), 2.59–2.37 (m, 5H), 2.30–2.14 (m, 3H), 2.11–2.01 (m, 1H), 1.86–1.61 (m, 7H), 1.60 (s, 3H), 1.59–1.50 (m, 4H), 1.42–1.21 (m, 12H), 0.87 (t, *J* = 6.9 Hz, 3H).

**^13^C NMR (100 MHz, CDCl_3_)** δ 177.3, 173.5, 153.4, 148.5, 147.3, 139.0, 137.0, 131.3, 126.5, 109.9, 108.2, 105.6 (2C), 101.5, 72.3, 71.0, 68.1, 60.9, 56.3 (2C), 45.7 (2C), 44.3, 39.9, 38.0, 35.5 (2C), 34.4, 34.3, 31.6, 30.6, 30.3, 29.2, 29.2, 29.2, 29.2, 28.2, 25.7, 24.9, 24.9, 23.7, 22.6, 20.0, 14.1.

**HRMS (ESI^+^):** *m/z* [M + Na]^+^ calcd. for C_53_H_66_O_12_Na, 917.4452; found, 917.4457.

${\left[\alpha \right]}_{D}^{25}$**:** -24.5 (*c* 1.07 in CHCl_3_).


**(2a*R*,4*S*,4a*S*,6*R*,9*S*,11*S*,12*S*,12b*S*)-9-(((*R*)-17-((*S*)-benzamido(phenyl)methyl)-2,2-dimethyl-6,15-dioxo-5,16-dioxa-10,11-dithia-2-silaoctadecan-18-oyl)oxy)-12-(benzoyloxy)-4,11-dihydroxy-4a,8,13,13-tetramethyl-5-oxo-3,4,4a,5,6,9,10,11,12,12a-decahydro-1H-7,11-methanocyclodeca[3,4] benzo[1,2-b]oxete-6,12b(2aH)-diyl diacetate (18b)**


Following the general procedure for the Steglich coupling, to a solution of carboxylic acid **16b** (36 mg, 0.11 mmol), EDC∙HCl (21 mg, 0.11 mmol), DMAP (6 mg, 0.045 mmol) in dry CH_2_Cl_2_ (1.5 mL, 0.05 M), **4** (75 mg, 0.09 mmol) is added to obtain 100 mg of product **18b** quantitatively. Reaction is monitored by TLC (4:6 *n*-hex/EtOAc) and purified by Biotage^®^ (gradient with *n*-hex/EtOAc eluent mixture).

**^1^H NMR (400 MHz, CDCl_3_)** δ 8.17–8.10 (m, 2H), 7.77–7.70 (m, 2H), 7.65–7.56 (m, 1H), 7.55–7.46 (m, 3H), 7.46–7.30 (m, 7H), 6.94 (d, *J* = 9.2 Hz, 1H), 6.32–6.21 (m, 2H), 5.98 (dd, *J* = 9.2, 3.1 Hz, 1H), 5.68 (d, *J* = 7.1 Hz, 1H), 5.51 (d, *J* = 3.1 Hz, 1H), 4.97 (dd, *J* = 9.6, 2.3 Hz, 1H), 4.44 (dd, *J* = 10.9, 6.6 Hz, 1H), 4.31 (d, *J* = 8.5 Hz, 1H), 4.20 (d, *J* = 8.5 Hz, 1H), 4.18–4.08 (m, 2H), 3.82 (d, *J* = 7.0 Hz, 1H), 2.75–2.61 (m, 4H), 2.61–2.48 (m, 4H), 2.46 (s, 3H), 2.42–2.32 (m, 3H), 2.24–2.12 (m, 4H), 2.09–1.96 (m, 4H), 1.94 (s, 3H), 1.92–1.80 (m, 2H), 1.68 (s, 3H), 1.23 (s, 3H), 1.13 (s, 3H), 1.03–0.91 (m, 2H), 0.03 (s, 9H).

**^13^C NMR (100 MHz, CDCl_3_)** δ 203.9, 171.2, 169.9, 168.1, 167.3, 167.0, 142.7, 137.0, 133.7, 132.1, 130.3 (2C), 129.2 (2C), 128.8 (2C), 128.6 (2C), 127.2 (2C), 126.6 (2C), 84.5, 76.5, 75.7, 75.2, 74.1, 72.1, 71.9, 62.8, 58.5, 52.8, 45.7, 37.7, 37.1, 35.7, 35.6, 32.8, 32.0, 26.9, 24.3, 24.0, 22.8, 22.2, 17.4, 14.9, 9.7, −1.4 (3C).

**HRMS (ESI^+^):** *m/z* [M + Na]^+^ calcd. for C_60_H_75_NO_17_S_2_SiNa, 1196.4143; found, 1196.4149.


**1-((1*S*,2*R*)-1-benzamido-3-(((2a*R*,4*S*,4a*S*,6*R*,9*S*,11*S*,12*S*,12b*S*)-6,12b-diacetoxy-12-(benzoyloxy)-4,11-dihydroxy-4a,8,13,13-tetramethyl-5-oxo-2a,3,4,4a,5,6,9,10,11,12,12a,12b-dodecahydro-1H-7,11-methanocyclodeca[3,4]benzo[1,2-b]oxet-9-yl)oxy)-3-oxo-1-phenylpropan-2-yl) 10-(2-(trimethylsilyl)ethyl) decanedioate (18a)**


Following the general procedure for the Steglich coupling, to a solution of carboxylic acid **16a** (71 mg, 0.071 mmol), EDC∙HCl (14.0 mg, 0.071 mmol), DMAP (4.0 mg, 0.030 mmol) in dry CH_2_Cl_2_ (1.5 mL, 0,05 M), **4** (50 mg, 0.059 mmol) is added to obtain 59.9 mg of product **18a** with 87% yield. Reaction is monitored by TLC (4:6 *n*-hex/EtOAc) and purified by Biotage^®^ (gradient with *n*-hex/EtOAc eluent mixture).

**^1^H NMR (400 MHz, CDCl_3_)** δ 8.10 (d, *J* = 7.4 Hz, 2H), 7.73 (d, *J* = 7.4 Hz, 2H), 7.58 (t, *J* = 7.4 Hz, 1H), 7.53–7.44 (m, 3H), 7.44–7.34 (m, 6H), 7.34–7.27 (m, 1H), 7.00 (d, *J* = 9.1 Hz, 1H), 6.28 (s, 1H), 6.27–6.16 (m, 1H), 5.93 (dd, *J* = 9.2, 3.5 Hz, 1H), 5.65 (d, *J* = 7.1 Hz, 1H), 5.50 (d, *J* = 3.5 Hz, 1H), 4.94 (dd, *J* = 9.6, 2.3 Hz, 1H), 4.42 (dd, *J* = 10.9, 6.6 Hz, 1H), 4.28 (d, *J* = 8.5 Hz, 1H), 4.21–4.09 (m, 3H), 3.79 (d, *J* = 7.0 Hz, 1H), 2.60–2.46 (m, 1H), 2.43 (s, 3H), 2.42–2.28 (m, 3H), 2.24 (t, *J* = 7.5 Hz, 3H), 2.19 (s, 3H), 2.17–2.07 (m, 1H), 1.92 (s, 3H), 1.90–1.76 (m, 1H), 1.65 (s, 3H), 1.63–1.50 (m, 4H), 1.32–1.18 (m, 11H), 1.11 (s, 3H), 1.01–0.91 (m, 2H), 0.02 (s, 9H).

**^13^C NMR (100 MHz, CDCl_3_)** δ 203.9, 174.0, 172.8, 171.2, 169.8, 168.2, 167.2, 167.0, 142.8, 137.1, 133.8, 133.7, 132.8, 132.0, 130.2 (2C), 129.3, 129.1 (2C), 128.8 (2C), 128.7 (2C), 128.5, 127.2 (2C), 126.7 (2C), 84.5, 81.1, 79.1, 76.5, 75.6, 75.2, 73.9, 72.1, 71.8, 62.4, 58.5, 53.0, 45.7, 43.2, 35.6, 34.5 (2C), 33.8, 29.1, 29.1, 29.0, 28.9, 26.8, 24.9, 24.7, 22.7, 22.2, 20.9, 17.3, 14.8, 9.7, −1.4 (3C).

**HRMS (ESI^+^):** *m/z* [M + Na]^+^ calcd. for C_62_H_79_NO_17_SiNa, 1160.5015; found, 1160.5019.


**1-((1*S*,2*R*)-1-benzamido-3-(((2a*R*,4*S*,4a*S*,6*R*,9*S*,11*S*,12*S*,12b*S*)-6,12b-diacetoxy-12-(benzoyloxy)-4,11-dihydroxy-4a,8,13,13-tetramethyl-5-oxo-2a,3,4,4a,5,6,9,10,11,12,12a,12b-dodecahydro-1*H*-7,11-methanocyclodeca[3,4]benzo[1,2-*b*]oxet-9-yl)oxy)-3-oxo-1-phenylpropan-2-yl) 10-(2,2,2-trichloroethyl) decanedioate (19a)**


Following the general procedure for the Steglich coupling, to a solution of carboxylic acid **18a** (24 mg, 0.071 mmol), EDC∙HCl (14 mg, 0.071 mmol), DMAP (4 mg, 0.030 mmol) in dry CH_2_Cl_2_ (1.5 mL, 0.05 M), **4** (50 mg, 0.059 mmol) is added to obtain 60 mg of product **19a** with 87% yield. Reaction is monitored by TLC (4:6 *n*-hex/EtOAc) and purified by Biotage^®^ (gradient with *n*-hex/EtOAc eluent mixture).

**^1^H NMR (400 MHz, CDCl_3_)** δ 8.18–8.10 (m, 2H), 7.78–7.70 (m, 2), 7.65–7.57 (m, 1H), 7.56–7.47 (m, 3H), 7.46–7.30 (m, 7H), 6.87 (d, *J* = 9.2 Hz, 1H), 6.32–6.21 (m, 2H), 5.95 (dd, *J* = 9.2, 3.2 Hz, 1H), 5.68 (d, *J* = 7.1 Hz, 1H), 5.50 (d, *J* = 3.2 Hz, 1H), 4.98 (dd, *J* = 9.8, 2.3 Hz, 1H), 4.74 (s, 2H), 4.45 (dd, *J* = 10.9, 6.6 Hz, 1H), 4.32 (d, *J* = 8.4 Hz, 1H), 4.20 (d, *J* = 8.4 Hz, 1H), 3.82 (d, *J* = 7.0 Hz, 1H), 2.56 (ddd, *J* = 14.7, 9.8, 6.6 Hz, 1H), 2.49–2.28 (m, 8H), 2.23 (s, 3H), 2.21–2.10 (m, 1H), 1.94 (d, *J* = 1.4 Hz, 3H), 1.89 (ddd, *J* = 14.6, 11.0, 2.4 Hz, 1H), 1.79–1.51 (m, 7H), 1.35–1.21 (m, 11H), 1.13 (s, 3H).

**^13^C NMR (100 MHz, CDCl_3_)** δ 203.9, 172.8 (2C), 171.3, 169.9, 168.2, 167.2, 167.1, 142.9, 137.1, 133.8, 133.8, 132.9, 132.1, 130.3, 129.3 (4C), 129.1 (4C), 128.8, 128.6, 127.2 (2C), 126.6 (2C), 95.2, 84.6, 81.2, 79.2, 76.5, 75.7, 75.2, 74.0, 73.9, 72.2, 71.9, 58.6, 52.9, 45.7, 43.3, 35.7, 35.6, 34.0, 33.8, 29.1, 29.0 (2C), 28.9, 26.9, 24.8, 22.8, 22.2, 20.9, 14.9, 14.3, 9.7.

**HRMS (ESI^+^):** *m/z* [M + Na]^+^ calcd. for C_59_H_68_Cl_3_NO_17_Na, 1190.3451; found, 1190.3454.


**1-((1*S*,2*R*)-1-benzamido-3-(((2a*R*,4*S*,4a*S*,6*R*,9*S*,11*S*,12*S*,12b*S*)-6,12b-diacetoxy-12-(benzoyloxy)-4,11-dihydroxy-4a,8,13,13-tetramethyl-5-oxo-2a,3,4,4a,5,6,9,10,11,12,12a,12b-dodecahydro-1*H*-7,11-methanocyclodeca[3,4]benzo[1,2-*b*]oxet-9-yl)oxy)-3-oxo-1-phenylpropan-2-yl) 10-(2,2,2-trichloroethyl) decanedioate (19b)**


Following the general procedure for the Steglich coupling, to a solution of carboxylic acid **18b** (26.0 mg, 0.071 mmol), EDC∙HCl (14.0 mg, 0.071 mmol), DMAP (4.0 mg, 0.030 mmol) in dry CH_2_Cl_2_ (1.5 mL, 0,05 M), **4** (50.0 mg, 0.059 mmol) is added to obtain 77.7 mg of product **19b** with 91% yield. Reaction is monitored by TLC (4:6 *n*-hex/EtOAc) and purified by Biotage^®^ (gradient with *n*-hex/EtOAc eluent mixture).

**^1^H NMR (400 MHz, CDCl_3_)** δ 8.17–8.10 (m, 2H), 7.78–7.69 (m, 2H), 7.65–7.55 (m, 1H), 7.56–7.46 (m, 3H), 7.47–7.31 (m, 7H), 6.90 (d, *J* = 9.2 Hz, 1H), 6.32–6.21 (m, 2H), 5.97 (dd, *J* = 9.2, 3.1 Hz, 1H), 5.68 (d, *J* = 7.1 Hz, 1H), 5.51 (d, *J* = 3.1 Hz, 1H), 4.97 (dd, *J* = 9.7, 2.3 Hz, 1H), 4.74 (s, 2H), 4.45 (dd, *J* = 10.9, 6.6 Hz, 1H), 4.32 (d, *J* = 8.4 Hz, 1H), 4.23–4.17 (m, 1H), 3.82 (d, *J* = 7.0 Hz, 1H), 2.74–2.46 (m, 9H), 2.46 (s, 3H), 2.38 (dd, *J* = 15.4, 9.3 Hz, 1H), 2.23 (s, 3H), 2.22–2.12 (m, 1H), 2.12–1.95 (m, 4H), 1.95 (s, 3H), 1.94–1.83 (m, 1H), 1.68 (s, 3H), 1.23 (s, 3H), 1.13 (s, 3H).

**^13^C NMR (100 MHz, CDCl_3_)** δ 203.9, 172.8 (2C), 171.3, 169.9, 168.2, 167.2, 167.1, 142.9, 137.1, 133.8, 133.8, 132.9, 132.1, 130.3, 129.3 (4C), 129.1 (4C), 128.8, 128.6, 127.2 (2C), 126.6 (2C), 95.2, 84.6, 81.2, 79.2, 76.5, 75.7, 75.2, 74.0, 73.9, 72.2, 71.9, 58.6, 52.9, 45.7, 43.3, 35.7, 35.6, 34.0, 33.8, 28.9, 26.9, 24.8, 22.8, 22.2, 20.9, 14.9, 14.3, 9.7.

**HRMS (ESI^+^):** *m/z* [M + Na]^+^ calcd. for C_57_H_64_Cl_3_NO_17_S_2_Na, 1226.2579; found, 1226.2583.


**10-(((1*S*,2*R*)-1-benzamido-3-(((2a*R*,4*S*,4a*S*,6*R*,9*S*,11*S*,12*S*,12b*S*)-6,12b-diacetoxy-12-(benzoyloxy)-4,11-dihydroxy-4a,8,13,13-tetramethyl-5-oxo-2a,3,4,4a,5,6,9,10,11,12,12a,12b-dodecahydro-1*H*-7,11-methanocyclodeca[3,4]benzo[1,2-*b*]oxet-9-yl)oxy)-3-oxo-1-phenylpropan-2-yl)oxy)-10-oxodecanoic acid (11a)**


Following the reported procedure described by Negretti et al. [32], to a solution of the trichloroethyl ester **19a** (59.9 mg, 0.051 mmol) in AcOH/MeOH 1:1 (2 mL, 0.03 M), zinc dust (83.4 mg, 1.275 mmol) is added at r.t. under vigorous stirring. The reaction is monitored by TLC (*n*-hex/EtOAc 4:6 + 1% HCOOH) and, after 4 h, is filtered over celite washing with MeOH. The organic phase is then washed with H_2_O extracting with abundant EtOAc, dried over anhydrous Na_2_SO_4_, filtered, and evaporated under reduced pressure. The crude mixture is purified by Biotage^®^ (gradient with *n*-hex/EtOAc + 1% HCOOH as eluent mixture), obtaining 30.1 mg of product **11a** with 57% yield.

**^1^H NMR (400 MHz, CDCl_3_)** δ 8.12 (d, *J* = 7.6 Hz, 2H), 7.72 (d, *J* = 7.6 Hz, 2H), 7.60 (t, *J* = 7.3 Hz, 1H), 7.50 (dd, *J* = 8.7, 6.6 Hz, 3H), 7.45–7.29 (m, 7H), 6.97 (d, *J* = 9.2 Hz, 1H), 6.30 (s, 1H), 6.25 (t, *J* = 9.1 Hz, 1H), 5.96 (dd, *J* = 9.2, 3.3 Hz, 1H), 5.68 (d, *J* = 7.1 Hz, 1H), 5.51 (d, *J* = 3.3 Hz, 1H), 4.97 (dd, *J* = 9.7, 2.3 Hz, 1H), 4.43 (dd, *J* = 10.9, 6.7 Hz, 1H), 4.30 (d, *J* = 8.5 Hz, 1H), 4.20 (d, *J* = 8.5 Hz, 1H), 3.81 (d, *J* = 7.0 Hz, 1H), 2.63–2.49 (m, 1H), 2.49–2.30 (m, 6H), 2.30–2.11 (m, 6H), 1.93 (s, 3H), 1.92–1.82 (m, 1H), 1.67 (s, 3H), 1.62–1.49 (m, 4H), 1.33–1.18 (m, 11H), 1.13 (s, 3H).

**^13^C NMR (100 MHz, CDCl_3_)** δ 203.9, 178.5, 172.9, 171.4, 170.0, 168.3, 167.6, 167.1, 142.8, 137.0, 133.8, 133.7, 133.0, 132.2, 130.3, 129.4 (4C), 129.2 (2C), 128.9, 128.8, 128.6, 127.2 (2C), 126.7 (2C), 84.6, 81.2, 79.1, 76.6, 75.8, 75.3, 73.9, 72.2, 71.9, 58.6, 53.0, 45.7, 43.3, 35.7, 33.9, 33.8, 29.0, 28.8, 26.9, 24.7, 24.7, 22.8, 22.2, 20.9, 14.9, 9.7.

**HRMS (ESI^+^):** *m/z* [M + Na]^+^ calcd. for C_57_H_67_NO_17_Na, 1060.4307; found, 1060.4311.


**10-(((1*S*,2*R*)-1-benzamido-3-(((2a*R*,4*S*,4a*S*,6*R*,9*S*,11*S*,12*S*,12b*S*)-6,12b-diacetoxy-12-(benzoyloxy)-4,11-dihydroxy-4a,8,13,13-tetramethyl-5-oxo-2a,3,4,4a,5,6,9,10,11,12,12a,12b-dodecahydro-1*H*-7,11-methanocyclodeca[3,4]benzo[1,2-*b*]oxet-9-yl)oxy)-3-oxo-1-phenylpropan-2-yl)oxy)-10-oxodecanoic acid (8a)**


Following the general procedure for the Steglich coupling, to a solution of carboxylic acid **11a** (30.1 mg, 0.029 mmol), EDC∙HCl (7.0 mg, 0.044 mmol), DMAP (2.0 mg, 0.015 mmol) in dry CH_2_Cl_2_ (1 mL, 0.05 M), **1** (14 mg, 0.044 mmol) is added to obtain 34.1 mg of product **8a** with 88% yield. Reaction is monitored by TLC (4:6 *n*-hex/EtOAc) and purified by Biotage^®^ (gradient with *n*-hex/EtOAc eluent mixture).

**^1^H NMR (400 MHz, CDCl_3_)** δ 8.13 (d, *J* = 7.7 Hz, 2H), 7.73 (d, *J* = 7.6 Hz, 2H), 7.61 (t, *J* = 7.4 Hz, 1H), 7.56–7.46 (m, 3H), 7.45–7.30 (m, 7H), 6.88 (d, *J* = 9.2 Hz, 1H), 6.54 (s, 1H), 6.38 (d, *J* = 1.7 Hz, 1H), 6.32–6.20 (m, 2H), 5.96 (dd, *J* = 9.2, 3.3 Hz, 1H), 5.68 (d, *J* = 7.1 Hz, 1H), 5.51 (d, *J* = 3.3 Hz, 2H), 4.97 (dd, *J* = 9.6, 2.3 Hz, 1H), 4.60 (s, 1H), 4.49–4.40 (m, 2H), 4.31 (d, *J* = 8.4 Hz, 1H), 4.20 (d, *J* = 8.5 Hz, 1H), 3.82 (d, *J* = 7.0 Hz, 1H), 3.48 (s, 1H), 2.64–2.31 (m, 11H), 2.22 (s, 3H), 2.20–1.98 (m, 3H), 1.95 (s, 3H), 1.93–1.70 (m, 6H), 1.68 (s, 3H), 1.63–1.52 (m, 7H), 1.42–1.24 (m, 12H), 1.23 (s, 3H), 1.13 (s, 3H), 0.87 (t, *J* = 6.7 Hz, 3H).

**^13^C NMR (100 MHz, CDCl_3_)** δ 204.0, 175.6, 172.8, 171.4, 171.3, 170.9, 169.9, 168.2, 167.2, 167.2, 142.9, 137.1, 135.1, 133.8, 132.9, 132.2, 130.4, 129.3 (4C), 129.2 (4C), 128.9, 128.6, 127.2 (2C), 126.6 (2C), 84.6, 81.2, 79.3, 76.6, 75.7, 75.2, 73.9, 72.3, 71.9, 58.6, 52.9, 45.7, 43.3, 38.0, 35.7, 35.6, 35.5, 34.4, 33.9, 31.6, 30.8, 30.6, 29.3, 29.2, 29.0, 26.9, 24.9, 24.8, 23.7, 22.8, 22.6, 22.3, 20.9, 14.9, 14.1, 9.7.

**HRMS (ESI^+^):** *m/z* [M + Na]^+^ calcd. for C_78_H_95_NO_18_Na, 1356.6447; found, 1356.6452.

${\left[\alpha \right]}_{D}^{25}$**:** -65.9 (*c* 0.69 in CHCl_3_).


**(1’*R*,2’*R*)-6-hydroxy-5’-methyl-4-pentyl-2’-(prop-1-en-2-yl)-1’,2’,3’,4’-tetrahydro-[1,1’-biphenyl]-2-yl 4-((4-oxo-4-(2-(trimethylsilyl)ethoxy)butyl)disulfaneyl)butanoate (20b)**


Following the general procedure for the Steglich coupling, to a solution of carboxylic acid **16b** (150 mg, 0.44 mmol), EDC∙HCl (93 mg, 0.049 mmol), DMAP (71 mg, 0.58 mmol) in dry CH_2_Cl_2_ (9 mL, 0.05 M), **1** (279 mg, 0.89 mmol) is added to obtain 159 mg of product **20b** with 57% yield. Reaction is monitored by TLC (9:1 *n*-hex/EtOAc) and purified by Biotage^®^ (gradient with *n*-hex/EtOAc eluent mixture).

**^1^H NMR (400 MHz, CDCl_3_)** 6.54 (bs, 1H), 6.38 (s, 1H), 5.98 (bs, 1H), 5.52 (bs, 1H), 4.65–4.55 (m, 1H), 4.44 (bs, 1H), 4.24–4.10 (m, 2H), 3.48 (bs, 1H), 2.78 (t, *J* = 7.1 Hz, 2H), 2.73 (t, *J* = 7.1 Hz, 2H), 2.68–2.58 (m, 2H), 2.53–2.38 (m, 5H), 2.29–2.07 (m, 4H), 2.07–1.98 (m, 2H), 1.87–1.67 (m, 5H), 1.61–1.51 (m, 5H), 1.38–1.21 (m, 5H), 1.03–0.95 (m, 2H), 0.87 (t, *J* = 6.8 Hz, 3H), 0.04 (s, 9H).

**^13^C NMR (100 MHz, CDCl_3_)** δ 172.9, 171.4, 156.8, 149.2, 143.1, 140.7, 139.1, 123.3, 114.8, 114.1, 111.6, 62.6, 45.7, 38.1, 37.6, 35.5, 32.5, 31.6, 30.6, 30.3, 29.8, 28.0, 24.1, 23.8, 22.6, 20.1, 17.3, 14.2, −1.5.

**HRMS (ESI^+^):** *m/z* [M + Na]^+^ calcd. for C_34_H_54_O_5_S_2_SiNa, 657.3080; found, 657.3084.


**4-((4-(((1’*R*,2’*R*)-6-hydroxy-5’-methyl-4-pentyl-2’-(prop-1-en-2-yl)-1’,2’,3’,4’-tetrahydro-[1,1’-biphenyl]-2-yl)oxy)-4-oxobutyl)disulfaneyl)butanoic acid (21b)**


To a well-stirred solution of compound **20b** (155.0 mg, 0.244 mmol) in dry THF (5.0 mL, 0.05M), 1M TBAF in THF (2.44 mL, 2.44 mmol) is added at 0 °C under nitrogen atmosphere. The reaction is stirred at r.t. and monitored by TLC (*n*-hex/EtOAc 7:3 + 1% HCOOH). The reaction is quenched with saturated NH_4_Cl aqueous solution and extracted with abundant EtOAc. The organic phase is then dried over anhydrous Na_2_SO_4_, filtered, and evaporated under reduced pressure. The crude mixture is then purified by flash chromatography (*n*-hex/EtOAc 8:2 + 1% HCOOH eluent mixture) to obtain 78 mg of the desired product **21b** with 65% yield.

**^1^H NMR (400 MHz, CDCl_3_)** δ 6.54 (bs, 1H), 6.38 (s, 1H), 5.98 (bs, 1H), 5.52 (bs, 1H), 4.60 (s, 1H), 4.44 (s, 1H), 3.48 (bs, 1H), 2.78 (t, *J* = 7.1 Hz, 2H), 2.73 (t, *J* = 7.1 Hz, 2H), 2.68–2.59 (m, 2H), 2.53–2.38 (m, 4H), 2.23–2.07 (m, 3H), 2.03 (p, *J* = 7.2 Hz, 2H), 1.86–1.66 (m, 5H), 1.62–1.52 (m, 5H), 1.35–1.24 (m, 5H), 0.93–0.82 (m, 3H).

**^13^C NMR (100 MHz, CDCl_3_)** δ 176.9, 171.4, 155.8, 149.2, 143.1, 140.7, 139.1, 123.3, 114.8, 114.1, 111.6, 45.7, 38.1, 37.6, 35.5, 32.5, 31.6, 30.6, 30.3, 29.8, 28.0, 24.1, 23.8, 22.6, 20.1, 14.2.

**HRMS (ESI^+^):** *m/z* [M + Na]^+^ calcd. for C_29_H_42_O_5_S_2_Na, 557.2371; found, 557.2377.

**(2a*R*,4*S*,4a*S*,6*R*,9*S*,11*S*,12*S*,12b*S*)-9-(((2*R*,3*S*)-3-benzamido-2-((4-((4-(((1’*R*,2’*R*)-6-hydroxy-5’-methyl-4-pentyl-2’-(prop-1-en-2-yl)-1’,2’,3’,4’-tetrahydro-[1,1’-biphenyl]-2-yl)oxy)-4-oxobutyl)disulfaneyl)butanoyl)oxy)-3-phenylpropanoyl)oxy)-12-(benzoyloxy)-4,11-dihydroxy-4a,8,13,13-tetramethyl-5-oxo-3,4,4a,5,6,9,10,11,12,12a-decahydro-1*H*-7,11-methanocyclodeca** [3,4]**benzo [1,2-*b*]oxete-6,12b(2a*H*)-diyl diacetate (8b)**

Following the general procedure for the Steglich coupling, to a solution of carboxylic acid **21b** (32 mg, 0.059 mmol), EDC∙HCl (11 mg, 0.059 mmol), DMAP (4 mg, 0.030 mmol) in dry CH_2_Cl_2_ (1.5 mL, 0.05 M), **4** (50 mg, 0.059 mmol) is added to obtain 49 mg of product **8b** with 61% yield. Reaction is monitored by TLC (4:6 *n*-hex/EtOAc) and purified by Biotage^®^ (gradient with *n*-hex/EtOAc eluent mixture).

**^1^H NMR (400 MHz, CDCl_3_)** δ 8.13 (d, *J* = 7.6 Hz, 2H), 7.76–7.70 (m, 1H), 7.64–7.55 (m, 1H), 7.55–7.45 (m, 3H), 7.45–7.30 (m, 7H), 6.93 (d, *J* = 9.2 Hz, 1H), 6.54 (s, 1H), 6.37 (d, *J* = 1.6 Hz, 1H), 6.32–6.20 (m, 2H), 5.97 (dd, *J* = 9.2, 3.2 Hz, 1H), 5.68 (d, *J* = 7.0 Hz, 1H), 5.51 (d, *J* = 3.2 Hz, 1H), 4.97 (d, *J* = 9.8 Hz, 1H), 4.61–4.56 (m, 1H), 4.48–4.39 (m, 2H), 4.31 (d, *J* = 8.4 Hz, 1H), 4.20 (d, *J* = 8.4 Hz, 1H), 3.81 (d, *J* = 7.0 Hz, 1H), 3.46 (bs, 1H), 2.76–2.43 (m, 15H), 2.37 (dd, *J* = 15.3, 9.4 Hz, 1H), 2.21 (s, 3H), 2.20–1.95 (m, 8H), 1.96–1.92 (m, 3H), 1.92–1.69 (m, 8H), 1.67 (s, 3H), 1.62–1.50 (m, 6H), 1.35–1.25 (m, 4H), 1.22 (s, 3H), 1.13 (s, 3H), 0.86 (t, *J* = 6.6 Hz, 3H).

**^13^C NMR (100 MHz, CDCl_3_)** δ 203.9, 172.0, 171.3, 169.9, 168.1, 167.2, 167.1, 142.8, 137.0, 133.8, 132.9, 132.1, 130.3, 129.3, 129.2, 128.8, 128.6, 127.2, 126.6, 117.2, 114.8, 84.6, 81.2, 79.2, 76.5, 75.7, 75.2, 74.2, 72.2, 72.0, 60.5, 58.6, 52.8, 45.7, 43.3, 38.1, 37.5, 37.1, 35.6, 35.5, 32.5, 32.0, 31.6, 30.6, 24.1, 24.0, 23.7, 22.8, 22.6, 22.2, 21.1, 20.9, 20.1, 14.9, 14.3, 14.1, 9.7.

**HRMS (ESI^+^):** *m/z* [M + Na]^+^ calcd. for C_76_H_91_NO_18_S_2_Na, 1392.5575; found, 1392.5579.

### 3.2. Dynamic Light Scattering (DLS)

DLS measurements were carried out by a 90 plus particle size analyzer (Brookhaven Instruments Corporation, Holtsville, NY, USA) equipped with a solid state He−Ne laser (wavelength = 661 nm). Experiments were carried out at a scattering angle of 90° on samples at 298 K. For both DLS and ζ-potential analysis, the purified samples were diluted in distilled water to a concentration of 200 µg/mL and briefly sonicated prior to the analysis. The results were expressed as mean ± standard deviation (SD) of three measurements.

### 3.3. Nanoparticle Preparation

Nanoparticle suspensions were prepared by a solvent displacement method [33]. Briefly, CBD containing analogs were dissolved in either ethanol (**6a,b**) or tetrahydrofuran (**7a,b** and **8a,b**) (4 mg/mL) and the solution was added dropwise to ultrapure water under stirring in order to have a final aqueous suspension 2 mg/mL. Finally, the organic solvent was evaporated under reduced pressure.

### 3.4. Cell Cultures

MSTO-211H (human biphasic mesothelioma) and MeT-5A (human mesothelial) cells were grown in RPMI-1640 (R6504, Sigma Chemical Co.) modified by the addition of 2.38 g/L Hepes, 0.11 g/L pyruvate sodium and 2.5 g/L glucose. HT-29 (human colorectal adenocarcinoma) and HepG2 (human hepatocellular carcinoma) cell lines were grown in RPMI-1640 (R6504, Sigma Chemical Co.) and MEM (M0894, Sigma Chemical Co.), respectively. 10% heat-inactivated fetal calf serum (Biowest), 100 U/mL penicillin, 100 μg/mL streptomycin and 0.25 μg/mL amphotericin B (Sigma Chemical Co.) were added to both media. The cells were cultured at 37 °C in a moist atmosphere of 5% carbon dioxide in air.

### 3.5. Inhibition Growth Assay

Cells (2.5–4 × 10^4^) were seeded into each well of a 24-well cell culture plate. After incubation for 24 h in standard conditions, various concentrations of the test nanoparticles or reference drug were added, and cells were then incubated for a further 72 h. Cells reached about 80% confluence in control condition. Untreated cells and cells treated with vehicle alone were also taken into consideration as controls. The trypan blue exclusion assay was performed to determine cell viability. Cytotoxicity data were expressed as GI_50_ values, that is, the concentration of the test agent inducing 50% reduction in cell number compared with control cultures.

### 3.6. Evaluation of Cell Death by Annexin V-FITC and Propidium Iodide Staining

The cell death was detected by a FITC Annexin V Apoptosis Detection Kit I (BD Pharmigen). MSTO-211H cells (2.5 × 10^5^) were seeded into each cell culture plate in complete growth medium. After incubation for 24 h, cells were treated with the test nanoparticle or the reference drug for a further 24 h so that in control condition about 50% confluence was reached. After treatment, cells were collected and resuspended in the supplied Binding Buffer at a density of at least 10^6^ cells/mL. Cell suspensions (500 μL) were added with Annexin V-FITC and propidium iodide (PI) and incubated for 15 min at room temperature in the dark, as indicated by the supplier’s instructions. The viable (Annexin V-negative/PI-negative), early apoptotic (Annexin V-positive/PI-negative), late apoptotic (Annexin V-positive/PI-positive) and necrotic (Annexin V-negative/PI-positive) cells were analyzed by FACSAria III flow cytometer and evaluated by FACSDiva software (BectonDickinson, Mountain View, CA, USA).

### 3.7. Confocal Microscopy Analysis

MSTO-211H cells (2 × 10^4^) were seeded on glass coverslips in 24-well plates and cultured until reached approximately 50% confluence. Cells were then incubated for a further 4 h in the presence of 100 μM tested nanoparticle or 1 μM paclitaxel, as reference drug. At the end of the experimental protocols, cells were washed twice with PBS, fixed with 4% formaldehyde for 15 min at room temperature, and permeabilized with 0.1% Triton X-100 in PBS for 5 min at room temperature. Then, cells were blocked by the incubation in 3% fetal bovine serum in PBS for 30 min at room temperature, and stained with the antibody conjugate Alexa Fluor 488 mouse anti-β-tubulin (BD Pharmingen) for 1 h at room temperature. The coverslips were mounted on glass slides by using Mowiol 40–88 (Sigma, St Louis, MO, USA) added with 1 μg/mL DAPI (Sigma-Aldrich). Images were acquired through a ×60 CFI Plan Apochromat Nikon objectives with a Nikon C1 confocal microscope and finally analyzed using NIS Elements software (Nikon Instruments, Florence, Italy), NIH Image J and Adobe Photoshop CS4 version 11.

## 4. Conclusions

A series of conjugates (**6–8A,B**) where CBD was used as a self-assembly inducer coupled with three different anticancer drugs was synthesized and characterized. These conjugates are able to self-assemble forming NPs, confirming the ability of CBD to induce this aggregation. The obtained NPs were characterized both for their physico-chemical properties and their biological activity. The ability to exert an antiproliferative effect was evaluated on three human tumor cell lines (MSTO-211H, HT-29, and HepG2), obtaining GI_50_ values in the low micromolar range. In particular, all the NPs containing 4,4′-dithiodibutyric acid as the linker, characterized by the presence of a disulfide bond (**6B**, **7B**, and **8B**), are remarkably more effective in inducing cell death with respect to the corresponding NPs presenting sebacic acid as linker, as predictable due to their easier intracellular cleavage. Further biological assays were carried out on MSTO-211H cells for the most effective NP **8B**, containing paclitaxel as the drug and 4,4′-dithiodibutyric acid as the linker, confirming the involvement of paclitaxel in cytotoxicity and cell death mechanism. This result supports the rationale of the approach, confirming the ability of the NP to address the drug inside the cell allowing its cytotoxic effect. In conclusion, these data further demonstrate the easy obtainment of self-assembled NPs by chemical functionalization of known anticancer drugs with a suitable self-assembly inducer, and of the possible modulation of their activity by varying the nature of the linker.

## Data Availability

Data are available contacting the corresponding authors upon reasonable request.

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
