# Peer review of "Cannabidiol as Self-Assembly Inducer for Anticancer Drug-Based Nanoparticles"

_molecules, 2022, doi:10.3390/molecules28010112_

Round 1

Reviewer 1 Report

The work deals with cannabidiol as a new self-assembly inducer in the formation of nanoparticles. The synthesis and characterization of the planned conjugates and their ability to form self-assembled NPs are reported. The ability to induce an antiproliferative effect was assayed on three human tumor cell lines (MSTO-211H, HT-29 and HepG2) and the maintenance of the cell target was assessed by confocal microscopy.

The work is interesting, rather well written, fits the scope of the journal, but should be revised to address my following raised points:

1) Please improve abstract to include specific results obtained, add some quantitative characteristics of NPs and their properties.

2) Introduction should be improved. Please include description of other nano-scaled self-assembled drug delivery system, alternative to conjugated. For example, those, using hydrophobic or electrostatic interactions for drug immobilization.

Possible references to look are:

- Caraway, C.A., et al. Polymeric Nanoparticles in Brain Cancer Therapy: A

Review of Current Approaches. Polymers 2022, 14, 2963.

- Berdiaki, A, et al. Assessment of Amphiphilic Poly-N-vinylpyrrolidone

Nanoparticles’ Biocompatibility with Endothelial Cells in Vitro and

Delivery of an Anti-Inammatory Drug. Mol. Pharm. 2020, 17, 4212–4225

- Sharma, A.K, et al. Overview of biopolymers as carriers of antiphlogistic

agents for treatment of diverse ocular inammations. Mater. Sci. Eng. C

2016, 67, 779–791.

3) As the work deals with plant extract reagent (CBD), its preparation, purification and characterization is crucial for the work results. This information on CBD must be included in materials and methods and to discussion in order to prove the quality and representation of the further data obtained.

4) What temperature was of DLS measuring and why? It is not mentioned.

5 )Microphotographs of NPs should be presented to show the morphology of prepared systems.

6) Positive and negative controls on cell growth was used? Please clarify.

Author Response

The work deals with cannabidiol as a new self-assembly inducer in the formation of nanoparticles. The synthesis and characterization of the planned conjugates and their ability to form self-assembled NPs are reported. The ability to induce an antiproliferative effect was assayed on three human tumor cell lines (MSTO-211H, HT-29 and HepG2) and the maintenance of the cell target was assessed by confocal microscopy.

The work is interesting, rather well written, fits the scope of the journal, but should be revised to address my following raised points:

1) Please improve abstract to include specific results obtained, add some quantitative characteristics of NPs and their properties.

The abstract was improved by adding further information on the nanoparticles added and on the results of their biological evaluation (lines 26-31, page 1).

2) Introduction should be improved. Please include description of other nano-scaled self-assembled drug delivery system, alternative to conjugated. For example, those, using hydrophobic or electrostatic interactions for drug immobilization.

Possible references to look are:

- Caraway, C.A., et al. Polymeric Nanoparticles in Brain Cancer Therapy: A

Review of Current Approaches. Polymers 2022, 14, 2963.

- Berdiaki, A, et al. Assessment of Amphiphilic Poly-N-vinylpyrrolidone

Nanoparticles’ Biocompatibility with Endothelial Cells in Vitro and

Delivery of an Anti-Inflammatory Drug. Mol. Pharm. 2020, 17, 4212–4225

- Sharma, A.K, et al. Overview of biopolymers as carriers of antiphlogistic

agents for treatment of diverse ocular inflammations. Mater. Sci. Eng. C

2016, 67, 779–791.

We thank the reviewer for suggesting these interesting references, but we don’t feel like they will be an appropriate addition in the introduction of this paper. Even though some of them refer to self-assembled nanosystems, their main characteristic is to be polymeric, which makes them significantly different form our drug-conjugate based nanoparticles. The focus of this work is the use of CBD – a small, biologically active compound – as self-assembly inducer, and for this purpose only nanoparticles having the same structure – i.e. formed by the covalent conjugation of small molecules - are cited in the introduction. We think that the discussion of other nanosystems – several others would have to be mentioned to cover all kinds of self-assembled drug delivery sistems published - will only lengthen the introduction and confuse the readers on the nature of the nanoparticles presented in the paper.

3) As the work deals with plant extract reagent (CBD), its preparation, purification and characterization is crucial for the work results. This information on CBD must be included in materials and methods and to discussion in order to prove the quality and representation of the further data obtained.

Information about CBD extraction, purification and characterization are now added in materials and method section (lines 318-326, page 13). Since this information are already known in literature and are not the focus of this work we don’t think it’s necessary to address them more in depth during in the discussion section.

4) What temperature was of DLS measuring and why? It is not mentioned.

The DLS was measured at 25 °C, that is the standard in our laboratories for this analysis. This information is now specified in the manuscript (line 759, page 21).

5 )Microphotographs of NPs should be presented to show the morphology of prepared systems.

Unfortunately we don’t possess images of the nanoparticles prepared, as their collection is not easy. This kind of nanoparticles in fact are almost completely transparent in TEM analysis since they lack heavy atoms that are necessary for detection, resulting in images of very poor quality.

6) Positive and negative controls on cell growth was used? Please clarify.

We thank the reviewer for the question. For each experiment we performed a control culture with untreated cells and a control culture with cells incubated in the presence of vehicle alone and both were used for the calculation of GI50 values. Moreover, for all tested cell lines positive controls were obtained by incubating cells with the reference drugs, i.e. paclitaxel, desacetylthiocolchicine, podophyllotoxin and CBD.

This is now clarified in the revised text (lines 197-199, page 9 and lines 788-790, page 22)

Reviewer 2 Report

- all schemes should be presented as figures

- vers 64 - sentence in the bracket

- what is the concentration of NP 8b in microscopy experiments? In the results we read about 100 uM of 8b but in material and methods section we have information about the concentration of 1 uM. By the way, what was the reason of usage such as concentrations of paclitaxel and NP 8b in these studies?  It should be concentration about GI50

- cell cultures - the Authors should add information about passaging of culture cells. What was the stage of growth curve of the cells used in the experiments?

- why the time of incubation with NP was different in case of inhibition growth assay (72 h) as compare to the evaluation of cells death (24 h)? I understand difference in the incubation in case of microscopy - thesestudies are very specific. However,  in  these experiments the time of incubation with the compounds shouldbe the same

- do the Authors have any suggestion why MSTO-211H cells were the most sensitive to 8b? It should be discussed

- the Authors should try to find the explanation/suggestion why studied NPs were not so effective as pure drugs. It should beincluded in discussion

- since anticancer drugs are cytotoxic also toward normal cells I suggest the Authors to perform their studies also on the non-cancer cells

Author Response

- all schemes should be presented as figures

As for the Molecules authors guidelines, we made a distinction between schemes and figures, where the schemes contain chemical reaction structures, differently from the figures. For clarity in distinguishing these two different types of images we don’t feel necessary to change the “schemes” diction into “figures”.

- vers 64 - sentence in the bracket

We apologize for the mistake in the figure reference, it was corrected.

- what is the concentration of NP 8b in microscopy experiments? In the results we read about 100 uM of 8b but in material and methods section we have information about the concentration of 1 uM. By the way, what was the reason of usage such as concentrations of paclitaxel and NP 8b in these studies?  It should be concentration about GI50

We apologize for the mistake in the Materials and methods section. The concentration of 8B in microscopy experiments is 100 microM, as indicated in the legend of Fig. 3. The mistake in Materials and methods section was corrected (line 810, page 22).

Such experimental conditions were chosen based on previous experiments that indicated a clear effect of paclitaxel already after 4 hours of treatment at 1 microM concentration. Starting from these data, the concentration of 8B was chosen maintaining the ratio obtained between the GI50paclitaxel/GI508B, about 1:100.

- cell cultures - the Authors should add information about passaging of culture cells. What was the stage of growth curve of the cells used in the experiments?

In cytotoxicity essays, cells reached approximately 80% confluence in control cultures, while in cell death experiments for control condition about 50% confluence is obtained. These information were added in the Materials and methods section of the revised version (page 22, lines 788-789 and 798-799)

- why the time of incubation with NP was different in case of inhibition growth assay (72 h) as compare to the evaluation of cells death (24 h)? I understand difference in the incubation in case of microscopy - thesestudies are very specific. However,  in  these experiments the time of incubation with the compounds shouldbe the same

The choice of these experimental conditions represents a standard approach in our laboratory for these studies, because the percentage of cells affected by the treatment with the drug (paclitaxel) allows to determine both an inhibitory and a stimulatory effect. A similar approach was published by us in ACS Med Chem Lett 2020, 11, 895-898 to investigate the effect on cell cycle of some drugs interacting with microtubule and NPs.

- do the Authors have any suggestion why MSTO-211H cells were the most sensitive to 8b? It should be discussed

The sensitivity of biphasic mesothelioma toward paclitaxel delivered by nanoparticles was already described and discussed in literature (J. Thorac. Cardiovasc. Surg, 2020, 160(3) e159-e168). In this paper the authors demonstrated that treatment with paclitaxel-loaded nanoparticles prolonged the survival of a murine model of malignant pleural mesothelioma obtained by intrathoracic injection of MSTO-211H cells. Our results appear in agreement with these data and support the use of NP for the delivery of the antitumor drug.

This is discussed in the revised version at page 10, lines 218-225)

Otherwise, the less sensitivity of colorectal cancer and hepatocellular carcinoma cell lines is not surprising, because drug resistance is a major hurdle for colorectal cancer treatment and hepatocellular carcinoma is classified as a highly chemoresistant disease (Biochimica et Biophysica Acta 1766 (2006) 184–196; BBA - Reviews on Cancer 1876 (2021) 188623; Chemotherapy 2012;58:381–386).

- the Authors should try to find the explanation/suggestion why studied NPs were not so effective as pure drugs. It should be included in discussion

The NPs to be effective should allow the release of the antitumor agent via a hydrolyzable linkage, like ester or disulphide moieties. Nevertheless, the efficiency of such reactions depends on many factors, including the availability of the involved intracellular enzymes and the kinetics of the release. It is then reasonable to assume that the amount of free drug in cytoplasm could be lower with respect to that obtained by incubating with the single agent, whose availability is not dependent and limited by the above factors.

This suggestion for the observed effects was included in the discussion of the revised version (page 9-10, lines 212-216)

- since anticancer drugs are cytotoxic also toward normal cells I suggest the Authors to perform their studies also on the non-cancer cells

We thank the reviewer for the correct suggestion and performed cytotoxicity assay by using Met-5A, non tumorigenic mesothelial cells, treated with paclitaxel and with the most active NP 8B. The obtained results are reported in the revised version along with a brief discussion (page 10, lines 236-245 and new refs 21 and 22) and the Materials and Methods section was modified accordingly (page 22, line 777).

Round 2

Reviewer 2 Report

The paper is ready for publication